# Biomarkers of Neurodegeneration in Post-Traumatic Stress Disorder: An Integrative Review

**DOI:** 10.3390/biomedicines11051465

**Published:** 2023-05-17

**Authors:** Ravi Philip Rajkumar

**Affiliations:** Department of Psychiatry, Jawaharlal Institute of Postgraduate Medical Education and Research (JIPMER), Puducherry 605006, India; jd0422@jipmer.ac.in; Tel.: +91-413-229-6280

**Keywords:** post-traumatic stress disorder, dementia, Parkinson’s disease, β-amyloid, inflammation, hippocampus, white matter integrity, genetics

## Abstract

Post-Traumatic Stress Disorder (PTSD) is a chronic psychiatric disorder that occurs following exposure to traumatic events. Recent evidence suggests that PTSD may be a risk factor for the development of subsequent neurodegenerative disorders, including Alzheimer’s dementia and Parkinson’s disease. Identification of biomarkers known to be associated with neurodegeneration in patients with PTSD would shed light on the pathophysiological mechanisms linking these disorders and would also help in the development of preventive strategies for neurodegenerative disorders in PTSD. With this background, the PubMed and Scopus databases were searched for studies designed to identify biomarkers that could be associated with an increased risk of neurodegenerative disorders in patients with PTSD. Out of a total of 342 citations retrieved, 29 studies were identified for inclusion in the review. The results of these studies suggest that biomarkers such as cerebral cortical thinning, disrupted white matter integrity, specific genetic polymorphisms, immune-inflammatory alterations, vitamin D deficiency, metabolic syndrome, and objectively documented parasomnias are significantly associated with PTSD and may predict an increased risk of subsequent neurodegenerative disorders. The biological mechanisms underlying these changes, and the interactions between them, are also explored. Though requiring replication, these findings highlight a number of biological pathways that plausibly link PTSD with neurodegenerative disorders and suggest potentially valuable avenues for prevention and early intervention.

## 1. Introduction

Post-Traumatic Stress Disorder (PTSD) is a chronic mental illness characterized by symptoms of increased vigilance and arousal, intrusive re-experiencing, avoidance behavior, and changes in effect and cognition lasting more than a month and following direct or indirect exposure to a traumatic stressor. Such stressors typically involve a risk of death, significant injury, or sexual violence [1,2]. It is estimated that PTSD affects around 5–10% of the world’s population, with an approximate 2:1 female-to-male ratio [3]. In countries and regions characterized by high levels of traumatic stress or civil unrest, the prevalence of PTSD is estimated to be much higher, affecting over 25% of the population [4]. Though the course of PTSD is heterogeneous, it tends to be chronic and persistent in many individuals. Standard treatments for PTSD include both psychological approaches, such as cognitive and group therapy, and pharmacological agents, such as selective serotonin reuptake inhibitors. However, a substantial proportion of patients do not respond fully to these treatments: short-term response rates have been estimated to be around 35–40% for psychotherapies [5] and 50–60% for pharmacotherapies [6]. A meta-analysis of remission rates for PTSD over a period of three or more years found that only 44% of patients could be considered “non-cases”; in other words, more than half remained symptomatic over this period [7]. PTSD is associated with significant levels of disability [8], impaired quality of life [9], high levels of comorbidity with psychiatric and substance use disorders [10,11], increased rates of several chronic medical illnesses [12,13], and an increased risk of suicide [14].

The processes involved in the pathogenesis of PTSD are complex, involving interactions between genetic vulnerability, exposure to stress or trauma in early life, and the nature and severity of the specific trauma exposure triggering the disorder [15,16,17,18,19,20]. These interactions may, in turn, be moderated by protective factors, which can be either individual or social [21]. Neuroimaging studies have identified several structural and functional alterations in the brains of individuals with PTSD, involving connections between cortical and limbic regions associated with cognition and emotion [22]. PTSD has also been associated with significant alterations in several key biochemical processes, particularly those related to hypothalamic–pituitary–adrenal (HPA) axis functioning, immune regulation, systemic inflammation, and oxidative stress [23,24,25]. As these changes may be associated with an increased risk of certain medical conditions, some researchers have labeled PTSD a systemic disorder [26]. More recent research has highlighted the fundamental cellular and electrophysiological changes associated with these disruptions [27,28,29,30,31], their behavioral correlates [32,33], and the mechanisms through which they influence the links between neural, immune, endocrine, and cardiovascular functioning [34]. This research has also identified novel pharmacological strategies for the prevention and management of PTSD, which may offer significant advantages over existing treatment approaches [35,36].

In addition to the comorbidities and complications listed above, several recent studies have suggested that PTSD may be a risk factor for the development of subsequent neurodegenerative disorders and, more specifically, for dementia. This association was initially documented in military veterans with a history of combat-related trauma [37]. It was subsequently found that PTSD was associated with an approximately 1.5 to 2-fold increase in the risk of dementia in both military and civilian populations [38], and this finding has been confirmed in large cohort studies involving both these groups [39,40]. Some researchers have also reported a longitudinal association between PTSD and the risk of Parkinson’s disease in both civilian and military populations [41,42]. These associations have led to speculations about the mechanisms linking PTSD with neurodegeneration, such as accelerated aging [43], increased systemic inflammation [44], and stress-related neurotoxicity affecting key brain regions [45]. However, other authors have emphasized the need for caution in assuming a direct causal link between PTSD and neurodegenerative disorders. These authors have pointed out a possible bi-directional relationship between PTSD and dementia [46] and the need to distinguish between correlation and causation, particularly in the presence of confounding factors [47].

In this context, the identification of specific biological markers known to be associated with neurodegeneration in patients with PTSD would be of significant value. Such markers would allow researchers to confirm the hypothesis of a causal relationship between PTSD and neurodegeneration, to identify the specific mechanisms mediating this association, and to develop preventive or early intervention strategies for patients who are in the pre-symptomatic or prodromal phase of neurodegeneration [48]. The aims of the current review are: (a)to summarize the existing research on biological markers associated with neurodegenerative disorders in individuals suffering from PTSD;(b)to critically examine the contributions of possible confounding factors;(c)to synthesize this information in a manner that would be useful to future researchers in this field.

## 2. Study Selection and Search Strategy

The current review was a scoping review of the existing research on biological markers associated with neurodegeneration in patients with a diagnosis of PTSD. For the purpose of this review, the definition of “biomarker” provided by the United States Food and Drug Administration-National Institutes of Health (FDA-NIH) Biomarker Working Group was used: “A defined characteristic that is measured as an indicator of normal biological processes, pathogenic processes or responses to an exposure or intervention”. Biomarkers “can be derived from molecular, histologic, radiographic, or physiologic characteristics” and are objective in nature, as opposed to clinical outcomes obtained through interviews or external observations of patients [49]. Given the heterogeneity of the available literature and the integrative nature of this review, all potential biomarkers were considered for inclusion. As a result, a formal systematic review or meta-analysis was not undertaken. Instead, a scoping review methodology was adopted in accordance with the PRISMA-ScR guidelines [50].

The criteria used to select studies for inclusion in this review were as follows:The study population should consist of patients with a diagnosis of PTSD, established using standard diagnostic criteria or rating scales, with or without a control or comparator group;Study participants should not currently fulfill the criteria for dementia or other neurodegenerative disorders;The study should measure one or more biomarkers that are actually or potentially linked to neurodegeneration, and this should be stated by the authors in the study methodology or protocol;Only human studies were included to ensure the specificity of any identified biomarkers for human subjects with PTSD;Only original research was included.

The PubMed and Scopus literature databases were searched using the key words “post-traumatic stress disorder” (OR its variants “post-traumatic stress disorder” and “PTSD”) AND either “neurodegeneration” (OR its variant “neurodegenerative”) OR “dementia” OR “Alzheimer’s disease (OR its variant “Alzheimer’s dementia”) OR “Parkinson’s disease”, AND various terms used to identify studies of biological markers: the broad terms “biological marker” and “biomarker” (OR their plural forms), OR “genetic” (OR variants) OR “immune” (OR variants) OR “inflammation” (OR variants) OR “amyloid” OR “tau protein” OR “endocrine” (OR variants such as “neuroendocrine”) OR “imaging” OR “MRI” OR “fMRI” OR “PET” OR “SPECT” OR “DTI” (OR their expansions, such as “magnetic resonance imaging” OR “positron emission tomography”). A complete list of the search strings used for the PubMed search, along with numerical results for the results retrieved, has been uploaded in Appendix A.

Through this process, a total of 342 citations were retrieved. After the removal of duplicate citations, the titles and abstracts of 252 citations were screened. Publication types other than original research in humans (*n* = 164) were excluded at this stage. In the next stage of the literature search, the full texts of the remaining 88 papers were checked to see if they fulfilled the review inclusion criteria. At this stage, 60 papers were excluded either because they did not measure biomarkers or because they did not include study participants with PTSD. Finally, the reference lists of the remaining 28 papers were searched for relevant research that might have been missed, and one further study was identified by this method. The current review thus covered a total of twenty-nine original studies of potential biomarkers of neurodegeneration in subjects diagnosed with PTSD [51,52,53,54,55,56,57,58,59,60,61,62,63,64,65,66,67,68,69,70,71,72,73,74,75,76,77,78,79].

The above process is depicted graphically in Figure 1.

## 3. Characteristics of the Included Studies

A complete list of the studies included in this review is provided in Table 1. From a preliminary review of each study’s methodology, it was found that they fell into five broad categories: brain imaging studies (*n* = 16), genetic, epigenetic, and gene expression studies (*n* = 8), biochemical marker studies (*n* = 9), immune-inflammatory marker studies (*n* = 4) and sleep-related marker studies (*n* = 5). The sum of these numbers is greater than 29 because twelve studies used more than one modality or examined the interaction between two or more biomarkers. Of the 29 studies included, 18 were conducted on military veterans and 11 on civilians; 4 of the civilian studies involved World Trade Center responders. Of the 18 studies involving military personnel, 8 studies included subjects with a history of mild to moderate traumatic brain injury (TBI) but with no diagnosis of dementia or other marked cognitive impairment related to their injury. All the studies that fulfilled the inclusion criteria for this review were from high-income countries in North America or Europe. Except for one study published in 2001, all the papers included in this review were published between 2014 and 2023.

## 4. Brain Imaging Studies

Both structural and functional imaging have been used to identify potential markers of neurodegeneration in subjects with PTSD. Structurally, PTSD has been associated with reduced volumes of specific cortical brain regions, including the rostral anterior cingulate cortex, insula, and right parahippocampal cortex [52,54]. Severe PTSD has been associated with a general reduction in overall cortical thickness, particularly in frontal regions [56] and, more specifically, in the right frontal lobe [70]. A longer duration of PTSD has been associated with reduced right hippocampal volume [51]. However, negative findings regarding an association between PTSD and hippocampal volume have also been reported by several researchers [52,69,74]. An overall reduction in structural gray matter connectivity between the prefrontal cortex and amygdala has also been observed in PTSD [52]. In contrast to these findings of volume loss, a single study has reported an increased amygdalar volume in veterans with PTSD, but only in those with a history of exposure to severe trauma [75].

Two studies have examined changes in white matter tract integrity in relation to PTSD. The first, which was conducted in veterans, found evidence of altered fractional anisotropy (FA) and diffusivity in the right cingulum, right inferior longitudinal fasciculus, and left anterior thalamic radiation; these changes were distinct from those associated with traumatic brain injury in the study sample [53]. The second, which involved civilians with trauma exposure related to the World Trade Center (WTC) terrorist attack, found reduced FA in the superior thalamic radiations and cerebellum in those with PTSD; in the subgroup of subjects with PTSD and mild cognitive impairment, further reductions in FA were reported in several other regions, including the fornix and right uncinate fasciculus [57]. Overall, structural imaging studies suggest that PTSD is associated with structural brain changes affecting cortical and limbic regions and white matter tracts, with some evidence of a right-sided predominance.

There are relatively few studies of functional imaging changes putatively linked to neurodegeneration in PTSD. In a study of civilians, increased variability of the BOLD signal, a measure of regional brain activity, was observed in patients with the dissociative subtype of PTSD compared to both other patients with PTSD and healthy controls [55]. A study of veterans did not report any specific functional MRI changes in patients with PTSD; however, two-thirds of the subjects also had a history of TBI [74]. A single study used positron emission tomography (PET) to estimate levels of amyloid beta and tau protein in veterans over a period of five years but did not find any significant association between PTSD and changes in these imaging markers [78].

## 5. Genetic, Epigenetic, and Gene Expression-Related Markers

Three association studies examined the associations between functional polymorphisms of specific genes and possible neurodegeneration related to PTSD. The first study examined the effects of functional polymorphisms of the *ALOX12* and *ALOX15* genes, which are related to oxidative stress, on PTSD-related reductions in brain volume. In this study, two specific single nucleotide polymorphisms (SNPs) of *ALOX12*—rs1042357 and rs10852889—appeared to mediate the association between PTSD symptom severity and reductions in right prefrontal cortical thickness [70]. The second study evaluated nine SNPs of the *BDNF* gene, a key regulator of neural plasticity, in relation to hippocampal volume changes in veterans with mild TBI and/or PTSD. In this study, a single SNP (rs1157659) interacted with TBI to reduce hippocampal volume; however, there was neither a direct effect of PTSD nor a genotype x PTSD effect on this outcome [74]. The third study examined the effects of two SNPs of the vitamin D-binding protein (*GC*) gene. In this study, PTSD was associated with lower serum vitamin D levels; of the two SNPs studied, homozygotes for the C allele of rs4588 were at a lower risk of PTSD, while carriers of the T allele of rs7041 were at a higher risk of this disorder [76].

In a study involving a small number of WTC responders with or without PTSD, peripheral blood gene expression was examined in four white cell subtypes. It was found that the expression of *FKBP5*, involved in hypothalamic-pituitary-axis functioning and the stress response, as well as the pseudogene *PI4KAP1*, were increased in all cell types in PTSD. Two further genes, *REST* and *SEPT4*, were upregulated in the monocytes of individuals with PTSD [58]. In a study of military veterans, an increased number of somatic mutations were observed in veterans with a diagnosis of PTSD. These mutations appeared to be linked to cytoskeletal and inflammation-related genes [59]. Finally, a post-mortem study of gene expression in the brains of veterans found four genes (*SNORA73B, COL6A3, GCNT1,* and *GPRIN3*) whose expression was associated with a lifetime diagnosis of PTSD [60].

To test for the possibility of functional interactions between the proteins encoded by these genes, all the above genes of interest were entered into the STRING database [80]. Two genes (*SNORA73B* and *PI4KAP1*) were excluded by the database as they did not encode any known functional product. Of the remaining nine genes analyzed, possible functional interactions were identified between three: *FKBP5*, *BDNF,* and *REST.* This is illustrated in Figure 2 below.

A single study examined the possible association between microRNAs (miRNAs) and PTSD. In this study, levels of 798 miRNAs were assessed. Of these, only miR-139-5p was significantly associated with PTSD symptom severity [77]. This particular miRNA is of significance as its expression has been found to differ significantly between patients with Alzheimer’s disease and healthy controls [81].

In addition to these results, a single study examined the association between PTSD and telomere length, a putative marker of cellular aging, in military personnel. In this study, PTSD was associated with a shortened telomere length only in veterans with a history of exposure to severe trauma; this finding was also associated with an increased volume of the amygdala and increased urinary norepinephrine [75].

## 6. Biochemical Marker Studies

The biomarkers most frequently studied in this context are those putatively associated with dementia, such as levels of amyloid-beta (Aβ) or tau protein. In a study involving 34 WTC responders, PTSD was associated with both lower plasma Aβ and a lower Aβ 42/40 ratio [61], while a larger study (*n* = 1173) involving WTC responders also found a lower Aβ 42/40 ratio in those with PTSD, as well as an association between this parameter and hippocampal volume reduction. However, in a study of trauma-exposed adults with and without PTSD, no significant differences in Aβ or tau levels were observed between groups [62]. An assay of extracellular vesicle levels of Aβ and tau in military veterans found no association between these markers and a diagnosis of PTSD [77]. A similar negative association between Aβ and tau levels and PTSD was observed in the cerebrospinal fluid of veterans, though 60% of these subjects also had a history of TBI [78]. Neurofilament light (NfL), a marker of neural axonal damage, was found to be elevated in patients with a history of PTSD and mild TBI in a single study [77]. However, two other studies did not observe a significant association between PTSD and NfL levels [61,79].

Among other biochemical markers possibly related to neurodegeneration, PTSD has been associated with reduced serum total vitamin D [76] and an increased risk of a metabolic syndrome characterized by elevated plasma glucose and dyslipidemia [75]. In the latter study, metabolic syndrome appeared to mediate the association between a diagnosis of PTSD and reduced volumes of specific regions of the frontal, temporal, and parietal cortices [75].

## 7. Immune and Inflammatory Marker Studies

Four studies have examined the relationship between PTSD and elevated levels of immune or inflammatory markers linked to neurodegeneration. In the earliest study in this population, CSF and plasma levels of interleukin-6 (IL-6) were examined in relation to levels of norepinephrine and adrenocortical hormones. CSF IL-6 was significantly higher in veterans with PTSD than in controls, and plasma IL-6 was positively correlated with norepinephrine only in PTSD cases [68]. In a study of individuals living near the WTC at the time of the 2001 attack, 43.2% fulfilled the criteria for PTSD. These individuals had significantly elevated levels of C-reactive protein (CRP), and CRP levels were positively correlated with PTSD symptom severity [63]. In contrast, two studies of military veterans yielded divergent results. In the first, PTSD severity was negatively correlated with IL-6 levels but positively associated with levels of soluble tumor necrosis factor-alpha receptor (sTNF-RII). sTNF-RII levels were negatively correlated with hippocampal volume, but this association was independent of PTSD diagnostic status [71]. In the second, no significant association was observed between PTSD and plasma levels of IL-10, IL-6, or TNF-α.

## 8. Sleep-Related Studies

Sleep-related behavioral and electrophysiological markers associated with neurodegeneration have been evaluated in relation to PTSD in five studies. Three of these involved military veterans; among these studies, two also included subjects with TBI. The first of these found an association between reduced sleep and reduced left hippocampal volume, but this was not related to a PTSD diagnosis [69]. The second, in which all participants had a history of mild TBI, found that PTSD was associated with self-reported sleep disturbances [64]. The third, which included only some subjects with TBI, found a significant association between PTSD and REM sleep behavior disorder (RSBD), considered a forerunner of neurodegeneration, both in subjects with and without TBI [65].

A polysomnographic study comparing patients with PTSD to those with idiopathic RSBD and healthy controls found a higher rate of REM sleep with atonia, as measured by electromyography, in those with PTSD [66]. A larger study based on self-report data, including over 20,000 adults in the context of the COVID-19 pandemic, found a significant positive association between self-reported PTSD symptoms and dream enactment behavior, which is a symptom of RSBD [67].

## 9. Integration of Existing Results on Biomarkers of Neurodegeneration in PTSD

A list of possible biomarkers associated with neurodegeneration in patients with PTSD, based on the existing literature, is provided in Table 2 below.

From the above results, it can be observed that even after discarding negative results, there are several potential biological markers of neurodegeneration in post-traumatic stress disorder. Neuroimaging studies have found that PTSD is associated with evidence of significant volume reductions in cortical regions—particularly in the right hemisphere—as well as reduced white matter integrity. In addition, there is some evidence to suggest that this disorder is associated with impaired grey matter structural connectivity between prefrontal and limbic regions. There is a significant overlap between these findings and those obtained in studies of patients with neurodegenerative disorders, particularly Alzheimer’s disease (AD). For example, progressive reductions in cortical thickness, implicating many of the same regions identified in patients with PTSD, have been identified both in patients with early evidence of Alzheimer’s pathology [82] and in patients with an established diagnosis of AD [83]. Similarly, disruptions of white matter integrity have also been documented across the spectrum of severity of AD [84,85]. Many of these alterations involve regions that have been identified in studies of individuals with PTSD, such as the fornix, uncinate fasciculus, and inferior longitudinal fasciculus [53,57]. Altered basal forebrain functioning has also been documented in both mild cognitive impairment and AD [86]; this is similar to findings reported in patients with the dissociative subtype of PTSD [55]. Though these associations are by themselves insufficient to establish a causal link, they suggest that PTSD may be associated with structural brain changes that differ from those of AD in degree rather than in kind.

Attempts to establish an association between biochemical markers of AD pathology, such as amyloid-beta and tau, have yielded inconsistent results in patients with PTSD. However, there is some evidence of indirect biochemical links between PTSD and neurodegenerative disorders. First, PTSD has been associated with an increased risk of metabolic syndrome, which is an established risk factor both for vascular dementia [87] and for disease onset and progression in AD [88,89]. Second, PTSD has been associated with low serum vitamin D, as well as with functional polymorphisms of the Gc protein that binds vitamin D and influences its levels. Vitamin D deficiency is associated with a modest but significant increase in the risk of dementia [90], as well as reductions in gray and white matter volumes [91], and there has been recent interest in vitamin D supplementation as a preventive measure against neurocognitive disorders [92]. Though both these findings require replication, they represent plausible pathways that could mediate the association between PTSD and neurodegeneration.

Immune and inflammatory mechanisms have been implicated in the progression of several neurodegenerative disorders, including AD [93], frontotemporal dementia [94], and Parkinson’s disease [95]. PTSD has been consistently associated with increases in certain markers of systemic inflammation [24,96], though it is not known to what extent these correlate with central nervous system inflammatory activity. In the current review, these findings were replicated in the context of possible links with neurodegeneration, with results specifically implicating CRP, IL-6, and soluble TNF-α receptor II. Of these three inflammatory markers, CRP has been associated with progression to dementia even after adjustment for confounding factors [97], while IL-6 has been specifically associated with vascular dementia [98].

Among the genetic markers associated with features of neurodegeneration in PTSD in this review, four are of particular interest, and there is evidence linking them to neurodegenerative disorders in three cases. Methylation of *ALOX12* has been associated with carotid artery intimal thickness, which is a risk factor for stroke and vascular dementia [99]. Septin-4, the protein encoded by the *SEPT4* gene, has been identified in the alpha-synuclein-positive cytoplasmic inclusion bodies seen in Parkinson’s and related diseases [100]. In a study comparing patients with AD and healthy controls, the expression of miR-139-5p was found to differ significantly between the two groups, suggesting a possible role for this specific microRNA in neurodegeneration [81,101]. Finally, on the basis of animal studies, it has been suggested that stress-induced changes in the methylation of *FKBP5* might contribute to late-life AD [102]. Thus, there are plausible mechanisms linking genetic markers associated with PTSD to specific types of neurodegenerative disorders.

REM sleep behavior disorder (RSBD) has been associated with an increase in the subsequent risk of neurodegenerative disorders, and more specifically of synucleinopathies such as Parkinson’s disease and dementia with Lewy bodies [103]. There is consistent evidence of an increase in RSBD, its symptoms, and its electrophysiological correlates in patients with PTSD [104]. Despite the consistency of this association, relatively little is known about the neurobiological mechanisms linking these two disorders and their implications for progression to a subsequent neurodegenerative disorder [105]. It has been suggested by some researchers that RSBD-like phenomena in PTSD have a distinctive pathophysiology characterized by hyperarousal and increased adrenergic activity rather than neurodegeneration [106]. If this hypothesis is correct, then symptoms of RSBD in PTSD may not necessarily increase the subsequent risk of dementia or Parkinson’s disease; however, it requires verification through both neurobiological and longitudinal clinical research.

The figure below (Figure 3) shows how these diverse findings can potentially be integrated. Innate genetic vulnerability, as well as epigenetic changes related to early life stressors and other environmental exposures, interact with exposure to one or more traumatic events, leading to the syndrome of PTSD. The biochemical, neurophysiological, and immune-inflammatory changes associated with PTSD can have an adverse impact on brain structure and function either directly or indirectly through atherogenesis and reduced cerebral blood flow. These changes may not be sufficient to lead to neurodegeneration in themselves, but they may interact with or amplify other risk factors, such as an innate genetic vulnerability towards neurodegenerative disorders, traumatic brain injury, or lifestyle factors. The outcomes of this process are likely to be heterogeneous, ranging from no or mild cognitive impairment to the clinical syndromes of AD, vascular dementia, or Parkinson’s disease.

A key question that arises in this regard is whether the associations depicted in Figure 3 are of a correlational or a causal nature. In other words, does the syndrome of PTSD itself lead to pathophysiological processes that increase the risk of neurodegeneration, or are both disorders related to an inherent vulnerability or diathesis? This question is difficult to answer based on current evidence. Some authors believe that there is at least provisional evidence of a causal link between PTSD and subsequent neurodegenerative disorders [43,45], while others favor a simple correlation based on shared genetic or environmental risk factors [44,47]. There is some evidence to support the latter view. For example, it has been found that increased systemic inflammation may precede PTSD and predispose to it rather than represent a consequence of this disorder [107]. Longitudinal studies of both civilian and military populations involving measurements of specific biomarkers prior to as well as following exposure to traumatic stress are required to answer this question. However, such studies may be difficult to conduct in the former group.

Another issue that arises in this context is the specificity of the association, whether correlational or causal, between PTSD and neurodegeneration. Is it non-specific in nature, applying to a wide range or spectrum of neurodegenerative disorders, or is it specific to certain conditions, such as AD? Alternately, are there distinct pathways (for example, oxidative stress-induced atherogenesis in vascular dementia or RSBD-like sleep pathologies in Parkinson’s disease) that mediate the association between PTSD and distinct types of neurodegenerative disorder? Research in this field is still in an early stage, and it is likely that more information on the specificity and consistency of these associations will be obtained in subsequent decades.

## 10. Limitations of the Existing Data

Despite the promising nature of the leads obtained through the current body of research, certain key limitations of existing studies deserve mention. First, the majority of studies have been conducted in Western countries, and it is not clear to what extent they can be generalized to other countries or cultures. Significant inter-ethnic and cross-national variations in the occurrence of PTSD following trauma exposure have been reported in large-scale studies. The reasons for these variations are not clear, but they may reflect innate differences in genetic vulnerability, as well as differences in environmental factors such as diet and social support [108,109]. Second, most of the published research on biomarkers has involved either military personnel or individuals affected by a single specific traumatic exposure, namely the WTC terrorist attacks. It is not clear if similar results would be obtained in individuals who develop PTSD in response to more commonly encountered forms of trauma, such as motor vehicle accidents or sexual assault [110]. Third, most positive findings reported in this review have not yet been replicated. Fourth, many studies of military veterans have included subjects with head trauma, which is an independent risk factor for dementia [111]. Fifth, the majority of studies have been cross-sectional in nature; therefore, it is not possible to draw clear inferences regarding a causal link between PTSD and neurodegenerative disorders based on their results. Finally, there is some evidence that the link between PTSD and certain neurodegenerative disorders may be non-specific and may extend to other stress-related disorders, such as acute stress disorder and adjustment disorder [40,112]. This facet requires further exploration in longitudinal studies.

## 11. Conclusions

Research on the links between PTSD and neurodegenerative disorders is a young and rapidly evolving field: 97% of the literature included in this review has been published in the last decade. There is significant evidence that several biological markers, including structural changes in cerebral gray and white matter, increased levels of pro-inflammatory markers, genetic polymorphisms related to oxidative stress and vitamin D levels, features of the metabolic syndrome, and RSBD-like parasomnias, may mediate the association between PTSD and neurodegenerative disorders, such as Alzheimer’s and Parkinson’s disease. It is possible to integrate these diverse findings into a biologically plausible framework. Though questions of causality and temporal sequence remain unsettled and certain methodological limitations need to be considered, these results are of importance from clinical, research, and public health perspectives. The current review can be considered complementary to existing reviews that emphasize the clinical and epidemiological links between PTSD and neurodegenerative disorders [46,113] and draws attention to the biological mechanisms and pathways that might link these conditions. Future research, ideally involving ethnically diverse populations, with a greater focus on civilians and an effort to minimize potential confounders, will hopefully clarify the true nature of the association between PTSD and these biological markers, as well as their value in predicting subsequent neurodegeneration and cognitive impairment. This would facilitate the development of interventions that target one or more of the biological pathways that these markers are associated with, hopefully leading to better pharmacological approaches to the prevention or early treatment of neurodegenerative disorders in those exposed to traumatic stress.

## Figures and Tables

**Figure 1 biomedicines-11-01465-f001:**
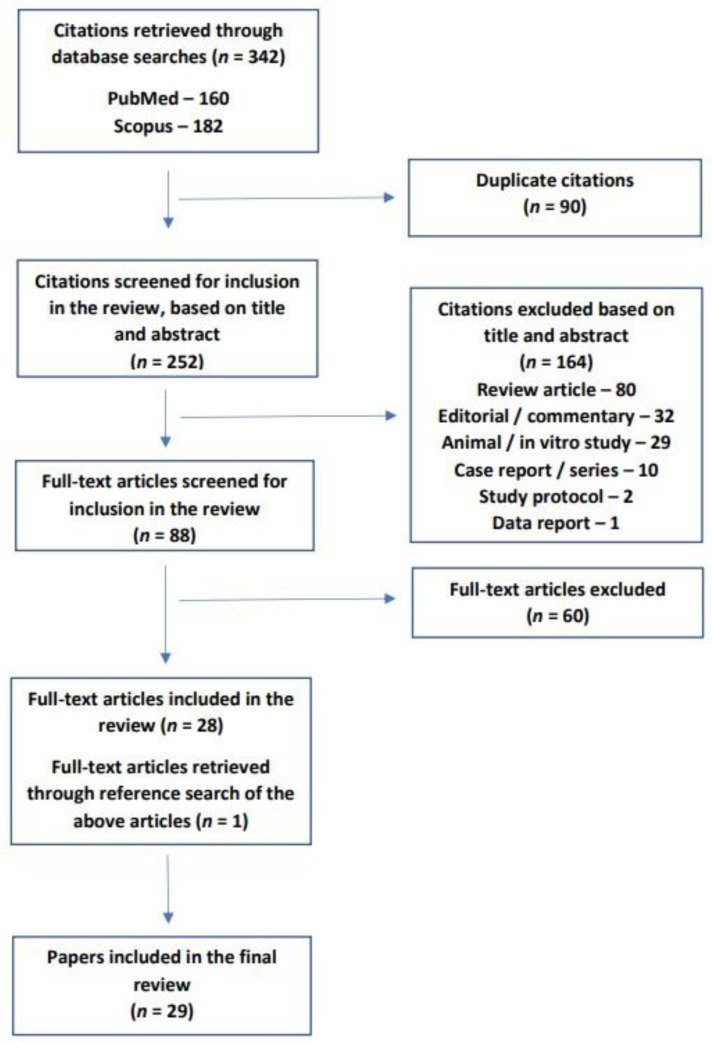
Flow diagram of the review process, based on PRISMA-ScR guidelines.

**Figure 2 biomedicines-11-01465-f002:**
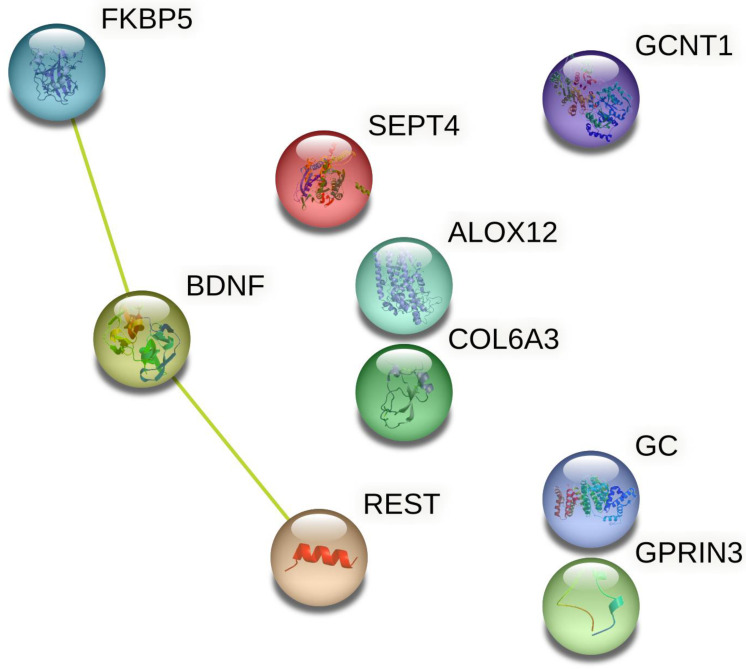
Protein–protein interactions for the products of genes linked to PTSD and neurodegeneration.

**Figure 3 biomedicines-11-01465-f003:**
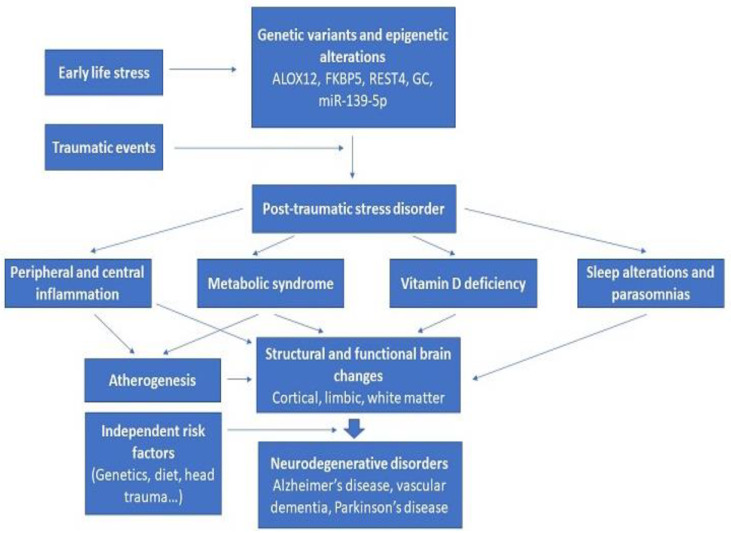
Integration of various risk factors linking post-traumatic stress disorder and neurodegenerative disorders.

**Table 1 biomedicines-11-01465-t001:** Details of all studies included in the current review.

Scheme	Modality	Study Population	Design	Biomarker(s) Studied	Results
**Brain imaging studies**					
Chao et al., 2014 [51]	Structural MRI	Military veterans with PTSD as per DSM-IV criteria (*n* = 55)	Cross-sectional; association	Hippocampal volume as test region; caudate nucleus volume as a control region	Duration of PTSD significantly and negatively correlated with right hippocampal volume, even after adjusting for confounders. No association between PTSD duration and left hippocampal or caudate volumes.
Mueller et al., 2015 [52]	Structural MRI	Military veterans with (*n* = 40) and without (*n* = 45) significant PTSD symptoms as measured using CAPS	Cross-sectional; case-control	Cortical and hippocampal volumes; structural connectivity of the prefrontal-limbic network	PTSD significantly associated with reduced rostral cingulate and insular cortical thickness but no hippocampal volume loss. Evidence of reduced prefrontal-limbic structural connectivity in PTSD.
Main et al., 2017 [53]	DTI	Military veterans (*n* = 109); 71.6% PTSD as per DSM-IV criteria; 57.8% mild TBI; 9.2% moderate TBI	Cross-sectional; association	FA and diffusivity of white matter fiber tracts	Altered parameters in left cingulum and inferior frontal-occipital fasciculus and right anterior thalamic tract specifically associated with TBI. Altered white matter parameters in right cingulum and inferior longitudinal fasciculus and left anterior thalamic radiation associated with PTSD.
Basavaraju et al., 2021 [54]	Structural MRI	Older adults (age ≥ 50 years) with a history of trauma exposure with (*n =* 55) and without (*n =* 36) PTSD	Cross-sectional; case-control	Cortical volume	Significant reduction of right parahippocampal cortical volume, but not other cortical regions, in PTSD.
Olivé et al., 2021 [55]	Functional MRI	Patients with PTSD (*n* = 103; 38 with dissociative subtype of PTSD); healthy controls (*n* = 46)	Cross-sectional; case-control	Variability of BOLD signal in basal forebrain regions	Increased BOLD signal variability in extended amygdala and nucleus accumbens in dissociative PTSD compared to both PTSD and controls.
Brown et al., 2022 [56]	Structural MRI	Military veterans (*n* = 254); 59.8% PTSD as per DSM-IV-TR criteria; 34.4% severe PTSD; (CAPS ≥ 60); 45.7% mild TBI	Longitudinal (2-year follow-up) with group comparisons (severe vs. non-severe PTSD)	Changes in cortical thickness, area, and volume	Severe PTSD associated with reduced cortical thickness, area, and volume, especially in frontal regions. More marked reductions in severe PTSD with mild TBI.
Kritikos et al., 2022 [57]	DTI	World Trade Center responders (*n* = 99); 48.4% cognitive impairment not amounting to dementia; 47.5% PTSD as per DSM-IV criteria	Cross-sectional; association	Whole-brain FA of white matter tracts	Reduced FA in fornix, cingulum, forceps minor, and right uncinate fasciculus in subjects with PTSD and cognitive impairment. Reduced FA in superior thalamic radiation and cerebellum in PTSD regardless of cognitive impairment.
**Genetic, epigenetic, and gene expression studies**					
Kuan et al., 2019 [58]	Peripheral blood transcriptome	World Trade Center responders with (*n* = 20) and without (*n* = 19) PTSD as per DSM-IV criteria	Cross-sectional; case-control	Transcriptome-wide analysis of gene expression in four peripheral blood immune cell subtypes	*FKBP5* and *PI4KAP1* upregulated across all cell types in PTSD. *REST* and *SEPT4* upregulated in monocytes in PTSD.
Sragovich et al., 2021 [59]	Somatic mutation	Military veterans with (*n* = 27) and without (*n* = 55) PTSD as per DSM-IV criteria	Cross-sectional; case-control	Rates of somatic mutations based on peripheral blood samples	Increased number of mutations related to cytoskeletal genes and inflammation in PTSD.
Wolf et al., 2021 [60]	Post-mortem gene expression	Post-mortem cortical brain tissue from military veterans (*n* = 97); 43.3% PTSD as per DSM-5 criteria; 30.9% alcohol use disorder	Post-mortem; association	DNA methylation-based estimates of cellular age in relation to chronological age (DNAm age residuals); gene expression in cortical tissue.	Specific interaction effects with age residuals identified for four genes (*SNORA73B, COL6A3, GCNT1,* and *GPRIN3*) specific to PTSD.
**Biochemical marker studies**					
Clouston et al., 2019 [61]	Plasma assay	World Trade Center responders with (*n* = 17) and without (*n* = 17) probable PTSD as per PCL-17	Cross-sectional; case-control	Plasma total amyloid-beta, amyloid-beta 42/40 ratio, total tau, and NfL	PTSD associated with lower plasma amyloid-beta and higher amyloid-beta 42/40 ratio.
Cimino et al., 2022 [62]	Serum assay	Adults, age ≥ 50, with a history of trauma exposure with (*n* = 44) and without (*n* = 26) subsequent PTSD as per DSM-5 criteria	Cross-sectional; case-control	Serum amyloid-beta 42 and 40 levels and ratio; serum total tau	No significant differences in amyloid-beta levels, ratios, or total tau levels between groups.
**Immune-inflammatory marker studies**					
Zhang et al., 2022 [63]	Serum assay	Residents living near the World Trade Center; 43.2% probable PTSD as per PCL; 50.3% dust cloud exposure	Cross-sectional; association	Serum CRP	Total CRP level and “high” CRP (>3 mg/L) both associated with PTSD. PCL score was a significant predictor of serum CRP.
**Sleep-related marker studies**					
Elliott et al., 2018 [64]	Polysomnography, self-report	Military veterans with a history of TBI (*n* = 130); 37.7% PTSD as per DSM-5 criteria	Cross-sectional; association	Sleep EEG/EMG; self-reported sleep disturbance; sensory (noise and light) sensitivity	Sleep disturbance and sensory sensitivity associated with PTSD in veterans with TBI.
Elliott et al., 2020 [65]	Polysomnography	Military veterans (*n* = 394); 28.6% probable PTSD as per PCL-5; 19.2% mild TBI	Cross-sectional; association	RSBD and other parasomnias	Increased rates of RSBD both in veterans with mild TBI and PTSD and in those with PTSD alone.
Feemster et al., 2022 [66]	Polysomnography	Patients with PTSD (*n* = 36), idiopathic RSBD (*n* = 18), and healthy controls (*n* = 51)	Cross-sectional; case-control	REM sleep with atonia	Higher REM sleep with atonia associated with PTSD independent of dream enactment behavior.
Liu et al., 2023 [67]	Self-report	Adults from fifteen countries (*n* = 21,870) during the COVID-19 pandemic; 3% COVID-19 positive; PTSD symptoms assessed using abbreviated PCL	Cross-sectional; association	Dream enactment behavior, weekly and lifetime	Screening positive for PTSD associated with a 1.2 to 1.4-fold increase in dream enactment behavior.
**Multi-modality studies**					
Baker et al., 2001 [68]	Plasma and CSF assays	Combat veterans (*n =* 11) with PTSD as per DSM-IV criteria; matched healthy controls (*n* = 8)	Cross-sectional; case-control	CSF levels of CRH, IL-6, norepinephrine; plasma levels of IL-6, ACTH, cortisol and norepinephrine	Increased CSF IL-6 in PTSD. Positive correlation between plasma IL-6 and norepinephrine in PTSD, but not in controls.
Mohlenhoff et al., 2014 [69]	Self-report (sleep); structural MRI (brain imaging)	Military veterans (*n =* 136); 7% PTSD as per DSM-IV criteria	Cross-sectional; association	Self-reported sleep disturbance; hippocampal volume	No association between PTSD and hippocampal volume. Possible association between sleep disturbance and left hippocampal volume.
Miller et al., 2015 [70]	Genetic association; structural MRI	Military veterans (*n* = 146); PTSD symptoms measured as a continuous variable using CAPS	Cross-sectional; interaction	Oxidative stress-related genes (*ALOX12* and *ALOX15*); prefrontal cortex thickness	PTSD symptom severity negatively correlated with right, but not left, prefrontal volume. Two SNPs of *ALOX12* moderated association between PTSD and right prefrontal volume.
O’Donovan et al., 2015 [71]	Serum assays; structural MRI	Military veterans with (*n* = 73) and without (*n* = 132) PTSD as per DSM-IV criteria	Cross-sectional; interaction	Serum IL-6 and sTNF-RII; hippocampal volume	sTNF-RII level negatively correlated with hippocampal volume regardless of PTSD diagnosis. PTSD severity associated with increased sTNF-RII and decreased IL-6.
Wolf et al., 2016 [72]	Plasma assays; structural MRI	Military veterans (*n* = 346); 77.2% PTSD as per DSM-IV criteria	Cross-sectional; interaction	Prevalence of metabolic syndrome as per NCEP-ATP III criteria; cortical thickness	Metabolic syndrome and its criteria more common in veterans with PTSD. Metabolic syndrome found to significantly mediate the association between PTSD and reduced cortical volume in precuneus, temporal cortex, rostral anterior cingulate cortex, and postcentral gyrus.
Hayes et al., 2017 [73]	Polygenic risk score; structural MRI	Military veterans (*n* = 160); 70% PTSD as per DSM-IV criteria; 65.6% mild TBI	Cross-sectional; interaction	Polygenic risk score for Alzheimer’s disease; cortical thickness	No significant association between cortical thickness and PTSD, either alone or in association with TBI or polygenic risk score.
Hayes et al., 2018 [74]	Genetic association; structural and functional MRI	Military veterans (*n* = 165); 66.7% mild TBI; 43% PTSD as per DSM-IV-TR criteria	Cross-sectional; interaction	*BDNF* genotype (9 SNPs); hippocampal volume; default mode network functional connectivity	No direct effect of PTSD on right or left hippocampal volume. TBI associated with reduced hippocampal volumes. Significant interaction between *BDNF* rs1157659 and TBI on hippocampal volume. No *BDNF* by PTSD interaction.
Kang et al., 2020 [75]	Peripheral blood and urine assays; structural MRI	Military veterans with (*n* = 102) and without (*n* = 113) PTSD as per DSM-IV criteria	Cross-sectional; interaction	Leukocyte telomere length; urinary catecholamines; amygdala volume	Shorter telomere length and increased amygdala volume associated with PTSD only in veterans exposed to high levels of trauma. Telomere shortening associated with increased urinary norepinephrine.
Terock et al., 2020 [76]	Genetic association; serum assay	1653 adults with a history of trauma exposure; 3.8% PTSD as per DSM-IV criteria	Cross-sectional; interaction	Serum total vitamin D; two specific SNPs of the *GC* gene	Lower serum vitamin D associated with PTSD. Vitamin D deficiency more frequent in those with PTSD. CC genotype of rs4588 associated with lower risk of PTSD. T allele of rs7041 associated with increased risk of PTSD.
Guedes et al., 2021 [77]	Extracellular vesicle assay	Military veterans (*n* = 144); 31.3% PTSD as per PCL-5 screening; 80.6% mild TBI	Cross-sectional; association	Extracellular vesicle levels of 798 miRNAs; extracellular vesicle and plasma levels of NfL, amyloid-beta 42 and 40, tau, IL-10, IL-6, TNF-α, and VEGF	Elevated extracellular vesicle levels of NfL in patients with mTBI and PTSD. Significant association between miR-139-5p and PTSD symptom severity.
Weiner et al., 2022 [78]	CSF assay; structural MRI; PET	Military veterans (*n =* 289); 60.6% PTSD as per DSM-IV criteria; 47.8% moderate to severe TBI	Longitudinal (5-year follow-up) with group comparisons	CSF amyloid-beta 42, total tau, and p-tau181; PET measures of amyloid-beta and tau; cortical, hippocampal, and amygdala volume	No significant association of PTSD with biochemical or imaging markers, either cross-sectionally or at follow-up
Kritikos et al., 2023 [79]	Plasma assay; structural MRI	World Trade Center responders (*n =* 1173); 11.2% probable PTSD as per PCL-C; 16.4% mild cognitive impairment; 4.3% possible dementia	Cross-sectional; interaction	Plasma amyloid-beta 40/42 ratio, p-tau 181, NfL; hippocampal volume (only in 75 participants)	Significant intercorrelation between amyloid-beta 40/42, p-tau181, and NfL. PTSD associated with elevated amyloid-beta 40/42 ratio. Amyloid-beta 40/42 and p-tau 181 associated with reduced hippocampal volume.

Note: Italics indicate gene names as per standard naming conventions. Abbreviations: ACTH, adrenocorticotrophic hormone; *ALOX12*, arachidonate 12-lipoxygenase gene; *ALOX15*, arachidonate 15-lipoxygenase gene; *BDNF*, brain-derived neurotrophic factor gene; BOLD, blood oxygen-level dependent; CAPS, Clinician-Administered PTSD Scale; *COL6A3*, collagen alpha-3 gene; CRH; corticotrophin-releasing hormone; CRP, C-reactive protein; CSF, cerebrospinal fluid; DSM, Diagnostic and Statistical Manual of Mental Disorders; DTI, diffusion tensor imaging; EEG, electroencephalogram; EMG; electromyogram; FA, fractional anisotropy; *FKBP5*, FK506 binding protein gene; GC, vitamin D-binding protein gene; *GCNT1*, glucosaminyl (N-acetyl) transferase 1 gene; *GPRIN3*, GPRIN family member 3 gene; IL, interleukin; miRNA, microRNA; MRI, magnetic resonance imaging; NCEP-ATP III, National Cholesterol Education Program—Adult Treatment Panel III; NfL, neurofilament light; PET, positron emission tomography; PCL, Post-traumatic Disorder Checklist; *PI4KAP1*, phosphatidylinositol 4-kinase alpha pseudogene 1; PTSD, post-traumatic stress disorder; REM, rapid-eye movement sleep; *REST*, RE1-silencing transcription factor gene; RSBD, REM sleep behaviour disorder; *SEPT4*, septin 4 gene; *SNORA73B*, small nucleolar RNA, H/ACA box 73B gene; SNP, single nucleotide polymorphism; TBI, traumatic brain injury; TNF, tumor necrosis factor; VEGF, vascular endothelial growth factor.

**Table 2 biomedicines-11-01465-t002:** Possible biomarkers linking post-traumatic stress disorder and neurodegeneration.

Biomarker Type	Level of Evidence
*Brain imaging*	
Reduced cortical thickness	++
Reduced volume of specific right cortical regions	++
Reduced white matter tract integrity	++
Reduced structural grey matter connectivity	+
Reduced right hippocampal volume	±
Increased amygdalar volume	?
Increased BOLD variability in the basal forebrain	?
*Genetic and epigenetic*	
Association between ALOX12 SNPs and reduced right prefrontal volume	+
Association between GC SNPs and PTSD risk	+
Upregulation of FKBP5, REST, SEPT4 in leukocytes	+
miR-139-5p	+
Reduced telomere length	?
Post-mortem expression of COL6A3, GCNT1, GPRIN3 in brain	?
*Biochemical*	
Reduced serum total vitamin D	+
Increased rate of metabolic syndrome	+
Elevated peripheral NfL	±
Reduced peripheral Aβ 42/40 ratio	±
*Immune-inflammatory*	
Serum CRP	+
Serum sTNF-RII	+
CSF IL-6	+
*Sleep-related*	
Increased rates of RSBD or dream enactment behavior	++
Increased REM sleep with atonia	+

Key: ++, positive evidence from more than one study; +, positive evidence from a single study; ±, conflicting results; ?, positive results only in sub-group analyses or results of uncertain significance.

## Data Availability

Not applicable.

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
