# Peer review of "Biomarkers of Neurodegeneration in Post-Traumatic Stress Disorder: An Integrative Review"

_biomedicines, 2023, doi:10.3390/biomedicines11051465_

Round 1
Reviewer 1 Report
Accumulating evidence suggests that PTSD is a risk factor for the development of neurodegenerative disorders. This review article introduces and integrates existing studies exploring biomarkers of neurodegenerative disorders in PTSD patients, which the author argues is useful for further verification of the hypothesis and the future research on the mechanism and clinical application of the association.
I evaluate that this article would attract readers’ attention to the relationship between PTSD and neurodegenerative disorders, and illuminate a direction for the future research on the issue. I only have minor comments as described below.
Individual comments
Line 35; Describe which gender is a majority.
Line 79; would -> with
Line 140; ‘and’ is duplicated.
Line 240; Is miR-139-5p involved in neurodegenerative disorders?
Line 248; ‘studied’ is duplicated.
Author Response
Accumulating evidence suggests that PTSD is a risk factor for the development of neurodegenerative disorders. This review article introduces and integrates existing studies exploring biomarkers of neurodegenerative disorders in PTSD patients, which the author argues is useful for further verification of the hypothesis and the future research on the mechanism and clinical application of the association.
I evaluate that this article would attract readers’ attention to the relationship between PTSD and neurodegenerative disorders, and illuminate a direction for the future research on the issue. I only have minor comments as described below.
Individual comments
1. Line 35; Describe which gender is a majority.
Response: I thank the reviewer for pointing out this omission and I apologize for the same. The sentence has been corrected as follows: "It is estimated that PTSD affects around 5-10% of the world’s population, with an approximate 2:1 female-to-male ratio."
2. Line 79; would -> with
Response: The sentence has been corrected as suggested.
Line 140; ‘and’ is duplicated.
Response: I apologize for this error. The duplicate "and" has been deleted.
Line 240; Is miR-139-5p involved in neurodegenerative disorders?
Response: I thank the reviewer for identifying this issue. The sentence has been corrected as follows: "A single study examined the possible association between microRNAs (miRNAs) and PTSD. In this study, levels of 798 miRNAs were assessed. Of these, only miR-139-5p was significantly associated with PTSD symptom severity [77]. This particular miRNA is of significance as its expression has been found to differ significantly between patients with Alzheimer’s disease and healthy controls [81]." Reference 81 is to a recent study that has identified altered expression of miR-139-5p in Alzheimer's dementia.
Line 248; ‘studied’ is duplicated.
Response: I apologize for this error. The duplicate "studied" has been deleted.
Reviewer 2 Report
This manuscript is a systematic review search in PubMed and Scopus for articles on Post-
Traumatic Stress Disorder (PTSD) and various biomarkers associated with
neurodegeneration. Not being a specialist in PTSD, I can not evaluate if it is exhaustive,
but many articles are described in good detail. Importantly, the author discusses deeply
the possible causative links versus a mere correlation.
My main criticism is in the description of the methodology at lines 112-121. Many
keywords are listed and it is presented as if AND was used to link them. I do not think this
is possible. Certainly, for the variants it should be OR, not AND. How could the author find
342 including all the keywords? He needs to be more precise about the algorithm in order
to be reproducible. Indeed, when I search for “Post-Traumatic Stress Disorder AND
neurodegeneration” in PubMed I only find 85 references. With PTSD AND
neurodegeneration, I find 102. Were the different keywords used independently and the
results summed up to give 342 citations? Also, “along with” at line 114 does mean AND? It
should be specified.
The review is very well written and clear. I only have minor editing corrections listed
below:
- Post-Traumatic Stress Disorder should be capitalized and used in the extended form
only the first time for each section followed by (PTSD). For example, at lines 95 and
101-102 it should be replaced by PTSD.
- Line 79: “would” replaced by “with”?
- Line 87: semicolon instead of comma
- Line 89; semicolon instead of “, and”
- Line 129: “in” replaced by “by”
- Line 142: “in” replaced by “on”
- Line 143: “in” replaced by “on”
- Line 248: “studied” repeated twice
- Line 251: comma between ) and “also and between “responders” and “also
- Line 273: remove comma
- Line 276: remove comma
- Line 282: remove comma
- Line 376: remove comma
- Line 381: remove second comma
- Line 403: replace semicolon with “or”
- Line 411: remove comma
- Line 439: remove comma
- Please, add a list of all the acronyms.
.
Author Response
This manuscript is a systematic review search in PubMed and Scopus for articles on Post-Traumatic Stress Disorder (PTSD) and various biomarkers associated with
neurodegeneration. Not being a specialist in PTSD, I can not evaluate if it is exhaustive, but many articles are described in good detail. Importantly, the author discusses deeply the possible causative links versus a mere correlation.
1. My main criticism is in the description of the methodology at lines 112-121. Many keywords are listed and it is presented as if AND was used to link them. I do not think this is possible. Certainly, for the variants it should be OR, not AND. How could the author find 342 including all the keywords? He needs to be more precise about the algorithm in order to be reproducible. Indeed, when I search for “Post-Traumatic Stress Disorder AND neurodegeneration” in PubMed I only find 85 references. With PTSD AND neurodegeneration, I find 102. Were the different keywords used independently and the results summed up to give 342 citations? Also, “along with” at line 114 does mean AND? It should be specified.
Response: I thank the reviewer for highlighting this limitation of the original manuscript. The section on "Search strategy" has been rewritten as follows:
"The PubMed and Scopus literature databases were searched using the key words “post-traumatic stress disorder” (OR its variants “posttraumatic stress disorder” and “PTSD”) AND either “neurodegeneration” (and its variant “neurodegenerative”) OR “dementia” OR “Alzheimer’s disease (and its variant “Alzheimer’s dementia”) OR “Parkinson’s disease”, AND various terms used to identify studies of biological markers: the broad terms “biological marker” and “biomarker” (and their plural forms), OR “genetic” (and variants) OR “immune” (and variants) OR “inflammation” (and variants) OR “amyloid” OR “tau protein” OR “endocrine” (and variants such as “neuroendocrine”) OR “imaging”OR “MRI” OR “fMRI” OR “PET” OR “SPECT” OR OR “DTI” (OR their expansions, such as “magnetic resonance imaging” and “positron emission tomography”). A complete list of the search strings used for the PubMed search, along with numerical results for the results retrieved, has been uploaded in Supplementary Table 1."
A new table (Supplementary Table 1) has been uploaded along with the revised manuscript.
The review is very well written and clear. I only have minor editing corrections listed below:
2. Post-Traumatic Stress Disorder should be capitalized and used in the extended form only the first time for each section followed by (PTSD). For example, at lines 95 and 101-102 it should be replaced by PTSD.
Response: I thank the reviewer for their positive and thoughtful comments regarding the manuscript. I have made corrections to the capitalization / abbreviation of PTSD as suggested above.
3. Line 79: “would” replaced by “with”?
Response: This has been replaced as suggested by the reviewer.
4. Line 87: semicolon instead of comma
- Line 89; semicolon instead of “, and”
Response: This has been replaced as suggested by the reviewer.
5. Line 129: “in” replaced by “by”
Response: This has been replaced as suggested by the reviewer.
6. Line 142: “in” replaced by “on”
- Line 143: “in” replaced by “on”
Response: This has been replaced as suggested by the reviewer.
7. Line 248: “studied” repeated twice
Response: This has been replaced as suggested by the reviewer.
8. Line 251: comma between ) and “also and between “responders” and “also
Response: This has been corrected as suggested by the reviewer.
9. Line 273: remove comma
- Line 276: remove comma
- Line 282: remove comma
- Line 376: remove comma
- Line 381: remove second comma
- Line 403: replace semicolon with “or”
- Line 411: remove comma
- Line 439: remove comma
Response: These have been corrected as suggested by the reviewer.
10. Please, add a list of all the acronyms.
Response: I apologize for this omission. As suggested by the reviewer, a complete list of all acronyms / abbreviations used in the text has been provided in the manuscript (Section 12).
Reviewer 3 Report
Major points:
1. Which additional information does this paper provide to the literature as there are recent studies on this topic:
Desmarais P, Weidman D, Wassef A, Bruneau MA, Friedland J, Bajsarowicz P, Thibodeau MP, Herrmann N, Nguyen QD. The Interplay Between Post-traumatic Stress Disorder and Dementia: A Systematic Review. Am J Geriatr Psychiatry. 2020 Jan;28(1):48-60 (study cited by the authors)
van Dongen DHE, Havermans D, Deckers K, Olff M, Verhey F, Sobczak S. A first insight into the clinical manifestation of posttraumatic stress disorder in dementia: a systematic literature review. Psychogeriatrics. 2022 Jul;22(4):509-520 (not cited by the authors)
2. Methods:
The search method needs specification:
Various key words were used in single or combined fashion.
Please indicate exactly the various combinations of the key words and also for each key word/combination the precise (numeric) results in findings.
3. Biomarker:
Please specify how you define biomarkers and if the proposed biomarkers in the literature fulfill the definition of biomarkers for neurodegenerative diseases.
Author Response
1. Which additional information does this paper provide to the literature as there are recent studies on this topic:
Desmarais P, Weidman D, Wassef A, Bruneau MA, Friedland J, Bajsarowicz P, Thibodeau MP, Herrmann N, Nguyen QD. The Interplay Between Post-traumatic Stress Disorder and Dementia: A Systematic Review. Am J Geriatr Psychiatry. 2020 Jan;28(1):48-60 (study cited by the authors)
van Dongen DHE, Havermans D, Deckers K, Olff M, Verhey F, Sobczak S. A first insight into the clinical manifestation of posttraumatic stress disorder in dementia: a systematic literature review. Psychogeriatrics. 2022 Jul;22(4):509-520 (not cited by the authors)
Response: I thank the reviewer for highlighting this issue. This has been discussed in the Conclusion (lines 471-474) as follows:
"The current review can be considered as complementary to existing reviews which emphasize the clinical and epidemiological links between PTSD and neurodegenerative disorders [46, 113], and draws attention to the biological mechanisms and pathways that might link these conditions."
The second paper suggested by the reviewer has also been cited in the revised manuscript (reference no. 113).
2. Methods:
The search method needs specification:
Various key words were used in single or combined fashion.
Please indicate exactly the various combinations of the key words and also for each key word/combination the precise (numeric) results in findings.
Response: I thank the reviewer for pointing out this limitation of the original manuscript. The "Search strategy" section has been rewritten as follows:
Complete numerical details have been provided for a sample search in Supplementary Table 1 which has been uploaded with the revised manuscript.
"The PubMed and Scopus literature databases were searched using the key words “post-traumatic stress disorder” (OR its variants “posttraumatic stress disorder” and “PTSD”) AND either “neurodegeneration” (and its variant “neurodegenerative”) OR “dementia” OR “Alzheimer’s disease (and its variant “Alzheimer’s dementia”) OR “Parkinson’s disease”, AND various terms used to identify studies of biological markers: the broad terms “biological marker” and “biomarker” (and their plural forms), OR “genetic” (and variants) OR “immune” (and variants) OR “inflammation” (and variants) OR “amyloid” OR “tau protein” OR “endocrine” (and variants such as “neuroendocrine”) OR “imaging”OR “MRI” OR “fMRI” OR “PET” OR “SPECT” OR OR “DTI” (OR their expansions, such as “magnetic resonance imaging” and “positron emission tomography”). A complete list of the search strings used for the PubMed search, along with numerical results for the results retrieved, has been uploaded in Supplementary Table 1."
3. Biomarker:
Please specify how you define biomarkers and if the proposed biomarkers in the literature fulfill the definition of biomarkers for neurodegenerative diseases.
Response: I thank the reviewer for raising this important concern. The following has been added to the Methodology as clarification:
"For the purpose of this review, the definition of “biomarker” provided by the United States Food and Drug Administration-National Institutes of Health (FDA-NIH) Biomarker Working Group was used: “A defined characteristic that is measured as an indicator of normal biological processes, pathogenic processes or responses to an exposure or intervention.” Biomarkers “can be derived from molecular, histologic, radiographic, or physiologic characteristics” and are objective in nature, as opposed to clinical outcomes which are obtained through interviews or external observations of patients [49]."
A citation to the original paper providing the definition of biomarkers and explaining the concept thereof has been added to the revised manuscript (reference no. 49).
Round 2
Author Response
I thank the reviewer for their valuable suggestions, and have made corrections as follows:
1. Line 28: remove comma before “and”.
Response: The comma has been removed.
2. Lines 131 and 132: OR in place of and before “its variants”.
Response: The word "and" has been replaced with "OR" as suggested.
3. Line 134: OR instead of and before “their plural forms”.
Response: The word "and" has been replaced with "OR" as suggested.
4. Lines 135, 136 and 138: OR instead of and before “variants”.
Response: The word "and" has been replaced with "OR" as suggested.
5. Line 297: remove comma before “and plasma”.
Response: The comma has been removed.
6. Line 306: remove comma before “or TNF”.
Response: The comma has been removed.
7. Line 410: remove comma before “or Parkinson’s”.
Response: The comma has been removed.
8. Line 469: remove comma before “and certain”
Response: The comma has been removed.
Reviewer 3 Report
The authors have replied and addressed all questions. No further comments from my side.
Author Response
I thank the reviewer for their kind comments on the manuscript and for their earlier suggestions, which have helped me to improve the paper significantly.